# Effects of Extraction Methods on the Characteristics, Physicochemical Properties and Sensory Quality of Collagen from Spent-Hens Bones

**DOI:** 10.3390/foods12010202

**Published:** 2023-01-03

**Authors:** Changwei Cao, Hailang Wang, Jinyan Zhang, Huan Kan, Yun Liu, Lei Guo, Huiquan Tong, Yinglong Wu, Changrong Ge

**Affiliations:** 1Department of Food Science and Engineering, College of Life Sciences, Southwest Forestry University, Kunming 650224, China; 2Graduate Department, Kunming University, Kunming 650214, China; 3College of Food Science, Sichuan Agricultural University, Ya’an 625014, China; 4College of Animal Science and Technology, Yunnan Agricultural University, Kunming 650201, China

**Keywords:** chicken bones, collagen, extraction method, characteristics, physicochemical properties

## Abstract

The present study used acetic acid, sodium hydroxide, and pepsin extract acid-soluble collagen (ASC), alkali-soluble collagen (ALSC), and pepsin-soluble collagen (PSC) from the bones of spent-hens, and the effects of three extraction methods on the characteristics, processing properties, antioxidant properties and acceptability of chicken bone collagen were compared. The results showed that the extraction rates of ASC, ALSC and PSC extracted from bones of spent-hens were 3.39%, 2.42% and 9.63%, respectively. The analysis of the amino acid composition, sodium dodecyl sulfate polyacrylamide gel electrophoresis (SDS-PAGE), Fourier transform infrared spectroscopy (FTIR), and ultraviolet full spectrum showed that the collagen extracted by the three methods had typical collagen characteristics and stable triple-helix structure, but the triple helical structure of PSC is more stable, and acid and alkaline extraction seems to have adverse effects on the secondary structure of chicken bone collagen. Differential scanning calorimetry (DSC) and scanning electron microscopy (SEM) scanning showed that PSC had higher thermal stability and more regular, loose, and porous microstructure. In addition, PSC has good processing properties, in vitro antioxidant activity, and organoleptic acceptability. Therefore, enzymatic hydrolysis was still one of the best methods to prepare collagen from bones of spent-hens, and enzyme-soluble collagen has wider application prospects in functional food and medicine and also provides an effective way for the high-value comprehensive utilization of waste chicken bone by-products.

## 1. Introduction

China is a major chicken-producing and -consuming country, with a chicken output of about 15 million tons in 2021 [1]. A large number of low-value spent-hens are often segmented and processed into meat paste, meat chest, meat floss, and other products [2,3], resulting in a large number of discarded chicken bones. Chicken bone accounts for about 25% of the total weight of a whole chicken and is an important source of animal protein and by-product of chicken processing. Dried chicken bones contain 12.0–35.0% protein, mostly collagen. In the past, due to the limitations of capital, technology, and other factors, a large number of chicken bones have not been fully utilized and were mostly processed into low-value-added products, such as feed, bone mud, bone meal, etc., or even discarded directly, which not only caused waste of resources but also environmental pollution [4,5,6]. With the rapid growth of laying hens breeding and the need to improve the utilization value of spent-hens, as well as people’s demand for sustainable and renewable high-quality protein, in the context of comprehensive utilization of resources, food security, and environmental principles, how to effectively use waste chicken bone resources in food production and processing is still one of the problems to be solved urgently. Therefore, the comprehensive and high-value utilization of by-products has attracted more and more attention and has become a hot topic of applied research in recent years [7,8].

Collagen is the main component of the animal extracellular matrix (ECM) and is the most abundant protein in bone, skin, and other by-products in the meat industry [9,10]. Animal tissues and organs are the main sources of collagen. The main feature of collagen is a stable triple-helix structure composed of three α polypeptide chains linked by hydrogen bonds, each polypeptide chain containing one or more regions characterized by repeated amino acid residues (Gly-X-Y); glycine (Gly) accounts for about 1/3 of all amino acids; X and Y are mainly proline and hydroxyproline [11]. The globular structure, spacing of triple helices, and variation in macromolecular length are the main criteria for distinguishing collagen types [12]. At least 29 types of collagen have been isolated and identified from animal tissues, most of which are type I, type II, and type III [7,13]. Collagen has been widely used in food, medicine, tissue engineering, cosmetics, and other fields because of its good biocompatibility, biodegradability, low antigenicity, and biological activity and has become a research hotspot [14,15].

Collagen exists in animal organs as an insoluble macromolecular structure and binds to proteoglycans, glycoproteins, etc. Therefore, appropriate extraction methods should be selected according to the differences in raw materials. Animal bones are an important source of collagen, but the hard tissue structure of bones makes collagen extraction more difficult. Therefore, establishing a stable, efficient, economical, and environmentally friendly extraction method is a key technical problem to be solved urgently in the deep processing of animal bones [16]. Although acid and enzymatic hydrolysis are the most commonly used methods for collagen extraction, in previous studies, different extraction methods or pretreatment methods have obvious effects on the characteristics and physical and chemical properties of collagen [17,18,19,20]. However, collagen is widely used because of its good functional activity, so it is necessary to analyze and compare the effects of different extraction methods on the extraction rate, physicochemical properties, and biological activity of chicken bone collagen.

Therefore, in this study, the discarded bones of spent-hens were used as raw materials, and acid-soluble, alkali-soluble, and enzyme-soluble chicken bone collagen was extracted from the bones of spent-hens by acetic acid, sodium hydroxide, and pepsin. Three collagens were identified and characterized, and the effects of different extraction methods on chicken bone collagen characteristics, physicochemical properties, antioxidant activity, and sensory acceptability were analyzed and compared for the first time.

## 2. Material and Method

### 2.1. Materials and Chemicals

The spent-hens (age 490 days) were provided by the practice chicken farm of Yunnan Agricultural University, which were separated, dried (at 65 °C for 12 h), and crushed (6-mesh sieve) to obtain chicken bone meal. The chemical reagents used were all analytical grade, purchased from Sichuan Xilong Science Co., Ltd. (Chengdu, China). The 8000 Da dialysis bag was purchased from Guangdong Yibo Biotechnology Co., Ltd. (Foshan, China); 4–20% denatured precast gel was purchased from Beijing Soleibo Biotechnology Co., Ltd. (Beijing, China). Pepsin (3000 U/g) was purchased from Shanghai Yeyuan Biotechnology Company (Shanghai, China). Diphenyl picryl phenyl hydrazine (DPPH), phenazine methyl sulfate (PMS), nitro blue tetrazolium (NBT), and nicotinamide adenine dinucleotide phosphate (NADH) were obtained from Shanghai McLean Biochemical Technology Co., Ltd. (Shanghai, China).

### 2.2. Sample Preparation

Pretreatment of chicken bone meal and extraction of collagen were performed according to the method described by Yu [21], Dai [22], Liu et al. [20], and us [23]. Sodium hydroxide, n-hexane, and EDTA were used for pretreatment to remove non-collagen, fat, and minerals from chicken bone meal. The pre-treated chicken bone meal was mixed with 0.5 M glacial aceticacid containing 0.1% (*w/v*) pepsin at a solid-to-liquid ratio of 1:10 (*w/v*) and continuously stirred and extracted at 4 °C for 48 h (pH 2.8). It was then filtered, and the filtrate was centrifuged at 15,000× *g* for 15 min at 4 °C. The pH of the supernatant was adjusted to 7.5–7.8 with NaOH solution, and NaCl was added to a final concentration of 1.5 M. After the mixture was kept undisturbed for 12 h at 4 °C for salting out, the collagen precipitate was centrifuged at 15,000× *g* for 15 min at 4 °C, then dissolved with 0.5 M acetic acid, dialyzed in pure water, freeze-dried, and the finished product was pepsin-soluble chicken bone collagen (PSC).

Acid-soluble chicken bone collagen (ASC) was directly extracted with 0.5 M glacial acetic acid according to the above method. The extraction of alkali-soluble chicken bone collagen (ALSC) according to the method described by Bi et al. [19] with slight modifications. Briefly, the pre-treated chicken bone meal was extracted with 0.12 M NaOH solution (pH 13.2) and then filtered. The filtrate was centrifuged and adjusted to pH 3.0 with acetic acid solution, and then ALSC was obtained by following the above steps.

### 2.3. Characterization of Chicken Bone Collagen

#### 2.3.1. Proximate Analysis

The contents of moisture, crude fat, crude protein, and ash in chicken bone meal, ASC, ALSC, and PSC were determined according to the methods AOAC 935.29, AOAC 920.39, AOAC 955.04, and AOAC 923.03, respectively, issued by the Association of Official Analytical Chemicals.

#### 2.3.2. Yield

The extraction rate of spent-hen bone collagen was calculated according to the weight of dried chicken bone powder (m_1_) and freeze-dried collagen (m_2_), using Equation (1)
(1)$$\mathrm{Yield}\;\left(\%\right)=\frac{m_{2}}{m_{1}}*100$$

#### 2.3.3. Sodium Dodecyl Sulfate Polyacrylamide Gel Electrophoresis (SDS-PAGE)

SDS-PAGE was performed according to the method described by Zhao et al. [17]. Collagen was suspended in 5% (*w/v*) SDS solution, mixed with an equal volume of 4× SDS-PAGE loading buffer, mixed with β-mercaptoethanol, water-bathed for 5 min, and centrifuged to take the supernatant for electrophoresis. The sample was loaded onto 4–20% precast gel, equilibrated at 80 V for 20 min, and then run with a constant voltage of 150 V for about 40 to 50 min, and proteins were stained with Coomassie brilliant blue R-250.

#### 2.3.4. Amino Acid Composition Analysis

The amino acid composition of chicken bone collagen samples was determined by the SykamS433D amino acid automatic analyzer (Munich, Germany) according to the method described by Liu et al. [20] and our [23] previous work.

#### 2.3.5. UV Absorption Spectrum Scanning

The ultraviolet absorption spectrum of collagen was determined with the method of Zhao et al. [17]. Collagen and acetic acid solution (0.5 M) were dissolved in a ratio of 1:1000 (*w/v*), and centrifuged, and the supernatant was collected and scanned with a UV spectrophotometer in the wavelength range of 190 to 400 nm, and the baseline was taken as the 0.5 M acetic acid solution.

#### 2.3.6. Fourier Transform Infrared Spectroscopy (FTIR)

The FTIR spectra of ASC, ALSC, and PSC were recorded using a FTIR spectrophotometer (MPA, Brukercompany, Karlsruhe, Germany) and according to the method described by Zhao et al. [17]. The IR spectra in the range of 4000 to 500 cm^−1^ with automatic signal gain were collected in 32 scans at a resolution of 4 cm^−1^.

#### 2.3.7. Thermal Stability Analysis

The melting temperature (Tm) of collagen was measured by differential scanning calorimetry (DSC) (DSC3500, NETZSCH Scientific Instruments Trading Ltd., Selb, Germany) according to the method of Akram and Zhang [24]. Briefly, the samples were placed in an aluminum crucible, and after rehydration and stabilization, the temperature was balanced at a rate of 5 °C/min and heated from 20 °C to 120 °C.

#### 2.3.8. Scanning Electron Microscopy (SEM)

According to the method of Zou et al. [25], the collagen was observed and photographed by SEM (Sigma 300, Carl Zeiss AG, Oberkochen, Germany). Freeze-dried collagen was cut into small pieces and fixed after spraying gold. Images were captured at 30× and 300× magnification with an accelerating voltage of 10 kV.

### 2.4. Processing Property Analysis

#### 2.4.1. Turbidity Analysis

The turbidity of collagen was measured according to the method reported by Ahmad et al. [26]. Briefly, the sample was configured as a solution (0.5%, *w/w*) with distilled water at 60 °C, and the absorbance of the solution was measured at 660 nm. The turbidity was calculated from the calibration curve of the kaolin standard solution, and the results were expressed in ppm.

#### 2.4.2. Soubility Analysis

Effect of pH on collagen solubility. The collagen samples were dissolved in 0.5 M acetic acid to obtain a final concentration of 3 mg/mL protein solution, and the pH of the solution was adjusted from 1 to 10. After centrifugation at 12,000× *g* for 20 min at 4 °C, the protein content of the supernatant was determined by the Bradford method [27].

Effect of NaCl on collagen solubility. The collagen was dissolved in 0.5 M acetic acid to obtain a protein solution with a final concentration of 6 mg/mL. A total of 5 mL of the solution was added to 5 mL acetic acid (0.5 M, NaCl was dissolved in the acetic acid in advance) to ensure that the NaCl content in the final solution ranged from 0 to 7%, and then the protein content of the supernatant was determined by the Bradford method [27].

#### 2.4.3. WHC and OAC

The water-holding capacity (WHC) of the samples was determined according to the method described by Nasrin et al. [28] with slight modifications. The samples were prepared in a 0.01% (*w/w*) solution with distilled water, and after a 37 °C water bath for 30 min, the supernatant was removed by centrifugation at 12,000× *g* for 20 min, and the absorbed water was weighed. WHC is expressed as the weight of water absorbed per gram of sample.

The oil absorption capacity (OAC) of collagen was determined according to the method described by Fan et al. [29] with slight modifications. Briefly, approximately 300 mg of collagen sample was dispersed in 20 mL of soybean oil, vortexed for 10 min at 25 °C, and left to stand at room temperature for 30 min. Then it was centrifuged at 12,000× *g* for 20 min and aspirated of free oil, and the volume was measured. The OAC value of collagen is expressed as the volume of oil absorbed per gram of sample.

#### 2.4.4. Emulsifying and Foaming Analysis

The emulsification activity index (EAI), emulsification stability index (ESI), foam expansion (FE), and foam stability (FS) of collagen were measured according to the method described by Akram and Zhang [24]. A 6 mL collagen solution (0.5%, *w/v*) was added to 2 mL soybean oil and homogenized at 10,000 r/min for 90 s. The emulsion was diluted 100 times with SDS (0.1%, *w/v*), and the absorbance of the emulsion was measured by spectrophotometer at 500 nm after standing for 0 min (A_0_) and 10 min (A_10_). EAI and ESI were calculated according to Equations (2) and (3).
(2)EAI m2g=2×2.303×A0×1000.5×0.001×0.25×10000
(3)$$\mathrm{ESI}\;\left(\min\right)=\frac{10\times A_{0}}{A_{0-}A_{10}}$$

Place 30 mL of collagen solution (2%, *w/v*) in a 100 mL graduated cylinder, adjust the pH of the solution to 7, and homogenize it at 10,000 r/min for 10 min. Record the volume of the solution before homogenization (*V*), after homogenization 0 min (*V*_0_), and after letting it stand for 60 min after homogenization (*V*_60_). FE and FS were calculated using Equations (4) and (5).
(4)$$\mathrm{FE}\;\left(\%\right)=\frac{V_{0}}{V}\;*\;100\;$$
(5)$$\mathrm{FS}\;\left(\%\right)=\frac{V_{60}}{V}\;*\;100$$

### 2.5. In Vitro Antioxidant Properties

The scavenging activities of ASC, ALSC, and PSC to DPPH•, HO•, O_2_^−^•, and ABTS^+^• radicals were determined according to the method reported by Li et al. [30]. Collagen samples were prepared in solutions with concentrations of 0.5, 1, 2, 3, 4, and 5 mg/mL to determine their antioxidant capacity. Concentration for 50% of maximal effect (EC50) was calculated according to the linear relationship between the free radical scavenging rate and the concentration of each sample. 

### 2.6. Color and Sensory Evaluation

The color of collagen samples was measured using a colorimeter following the methods described by Ahmad et al. [26] and the *L** *a** *b** chromatic aberration system. Before each measurement, the colorimeter was calibrated with a standard colorimetric plate. At least six points were measured per sample. The color of collagen samples is expressed as the mean value of lightness value (*L**), redness value (*a**), and yellowness value (*b**).

The sensory properties of collagen were evaluated according to the method described by Boran et al. [31]. The sample and deionized water were prepared in a 6.67% solution (*w/v*) in a test tube and evaluated after being water-bathed at 60 °C for 30 min. The sensory evaluation panel consisted of nine food graduate students with sensory research experience (4 men and 5 women, ages 23 to 26), and panelists were asked to assess the color, odor, aroma, viscosity, and transparency of the collagen solution using a 10-point quantitative scale to assess specific sensory attributes.

### 2.7. Statistical Analysis

All experiments were performed in at least 3 replicates, and results were expressed as mean ± standard deviation (SD). All results were analyzed using the SPSS 21 (IBM Inc., Armonk, NY, USA) software through one-way analysis of variance (ANOVA) and Duncan’s multiple range test, with the significance level set to *p* < 0.05. Origin Pro 2017 (Origin Lab, Northampton, MA, USA) was used to plot the data.

## 3. Result and Analysis

### 3.1. Characterization of Chicken Bone Collagen

#### 3.1.1. Yield and Chemical Composition Analysis

The extraction rate of collagen and the chemical composition of collagen and raw materials are shown in Table 1. As shown in Table 1, the extraction method had a significant effect on the yield of chicken bone collagen (*p* < 0.05), with the highest yield of PSC and the lowest of ALSC. In addition, the moisture, ash, protein, and fat contents of chicken bone collagen were similar to those reported by Akram [24]. Moisture and ash content were no significantly different (*p* > 0.05); crude protein in PSC was significantly higher than in ASC and ALSC (*p* < 0.05), and there was no significant difference between ASC and ALSC (*p* > 0.05); crude fat was significantly different among the three collagens (*p* < 0.05), with PSC being the lowest and ALSC being the highest.

#### 3.1.2. Amino Acid Composition Analysis

As shown in Table 2, the amino acid compositions of ASC, ALSC, and PSC are similar and all have the characteristics of collagen. The main manifestation is that Gly is the main amino acid in ASC, ALSC, and PSC, accounting for nearly 1/3 of the total amino acids (respectively: 26.74%, 22.67%, and 27.86%); hydroxyproline is a special amino acid in collagen, accounting for 9.51%, 6.29%, and 9.83%, respectively. In general, the difference in amino acid content between ASC and PSC was small, and the difference between ALSC with ASC and PSC was obvious.

#### 3.1.3. SDS-PAGE

Figure 1A shows the SDS-PAGE electrophoresis bands of ASC, ALSC, and PSC, which are similar to chicken bone gel protein [32], chicken breast cartilage collagen [24], and chicken feet skin collagen [33]. As shown in Figure 1A, PSC was composed of α, β, and γ chains, and ALSC had γ and α chains, while ASC only has an α chain, which may be related to the destruction of the collagen structure by acid and alkaline extraction. The α chain consists of α_1_ and α_2_ chains located between 100 and 135 kDa; the β chain located near 180 kDa is a dimer of an α chain; the γ chain located near 245 kDa is a trimer of an α chain. PSC has a low-molecular-weight band between 35 and 48 kDa, which may be caused by enzyme cleavage of collagen terminal peptide [34] or the further degradation of collagen by pepsin. According to previous reports, this spent-hen bone collagen may be type II collagen [35].

#### 3.1.4. UV

The absorption of ultraviolet light by chromogenic groups in protein molecules produces ultraviolet absorption spectrum. The triple-helix structure and the C=O, -COOH, and other perssad of collagen ensure it has a maximum absorption peak in the range of 210 to 240 nm [17,36]. As shown in Figure 1B, the UV absorption spectra of PSC, ASC, and ALSC have a maximum absorption peak at 215.77 nm, 207.12 nm, and 206.39 nm, respectively, while there is no obvious absorption peak at other wavelength segments (190–400 nm).

### 3.2. FTIR

The FTIR spectra (4000–500 cm^−1^) and characteristic peak information (amide A, B, I, II, and III) of ASC, ALSC, and PSC are shown in Figure 2 and Table 3. Characteristic peaks represent high polar bonds or functional groups in collagen. As shown in Figure 2, the main absorption bands of the three samples are located in the amide band region, including amide A peak (3400–3440 cm^−1^), amide B peak, amide I (1700–1600 cm^−1^), amide II (1550–1600 cm^−1^), and amide III (1300–1200 cm^−1^) [33].

The amide A band is closely associated with N-H stretching and hydrogen bonding (Table 3). Usually, the wavenumber of N-H radical stretching vibration is in the range of 3400 to 3440 cm^−1^, but if the N-H radical participates in the formation of hydrogen bonds, the wavenumber will move to a lower frequency [17]. The wavenumber of amide A of ASC, ALSC, and PSC were 3340.91 cm^−1^, 3371.93 cm^−1^, and 3290.62 cm^−1^, respectively, indicating that some N-H radicals in ASC, ALSC, and PSC participated in the formation of hydrogen bonds, and the order of hydrogen bond formation was PSC > ALSC > ASC. The amide B band is related to the asymmetric tensile vibration of -NH_3_^+^ and CH_2_, while the movement of amide B to higher wavenumbers is related to the increase of CH_2_ and N-terminal free NH-NH_3_^+^ radicals [17,37]. The wavenumbers of amide B band of ASC, ALSC, and PSC are 2923.13 cm^−1^, 2923.51 cm^−1^, and 2921.42 cm^−1^, respectively, indicating that free NH_3_^+^ or CH_2_ in PSC is lower than that in ASC and ALSC.

Amide I bands and amide II bands, which are related to molecular order and the triple helical structure of collagen, are known as fingerprints of protein secondary structures and are generated by C=O stretching, N-H bending, and C-H stretching, respectively [38]. The amide I band is associated with C=O stretching vibrations along the polypeptide backbone or with COO-coupled hydrogen bonds and has a strong absorbance in the range of 1600 to 1700 cm^−1^. The amide I bands of ASC, AlSC, and PSC are 1644.51 cm^−1^, 1652.18 cm^−1^, and 1644.40 cm^−1^, respectively. The amide II band mainly reflects NH bending and CN tensile vibration, which usually occurs in the range of 1550 to 1600 cm^−1^. In other words, the amide II band specifies the number of NH radicals that form hydrogen bonds with the adjacent α-chain. Thus, the lower wavenumber of the amide II band indicates both an increased number of hydrogen bonds by the NH radical and a higher structural order [39]. Amide II bands of ASC, AlSC, and PSC are 1559.72 cm^−1^, 1560.49 cm^−1^, and 1548.30 cm^−1^, respectively. Yousefi et al. [39] suggested that amide II bands of collagen located at low wavenumbers have more hydrogen bonds between adjacent α-chains. This evidence suggests that there are more hydrogen bonds in PSC. The absorption of amide III band is related to the triple helical structure of collagen, including C-N stretching, planar bending of C-H amide bonds, and the vibration of CH_2_ radical on glycine main chain and proline side chain [40], and is typically located in the wavelength range of 1300 cm^−1^ to 1200 cm^−1^. Amide III bands of ASC, AlSC, and PSC were 1244.24 cm^−1^, 1250.84 cm^−1^, and 1243.22 cm^−1^, respectively, indicating that there were stable triple helical structures in the three collagens. In addition, the three collagens had transmission peaks between 1400 cm^−1^ and 1465 cm^−1^, which may be the pyrrole ring vibration of proline and hydroxyproline in collagen [40]. It has been reported that the intensity ratio of the amide II band and the 1450 cm^−1^ band was used to elucidate the collagen triple-helix structure of collagen, and the transmission ratio of the amide III band to the 1400–1465 cm^−1^ band is close to 1.0; the integrity of the triple-helix structure of the three types of collagens was confirmed [17,41]. Therefore, the complete triple-helix structure was still maintained in the ASC, ALSC, and PSC prepared in this study.

### 3.3. Thermal Stability

The thermal stability of ASC, ALSC, and PSC are shown in Figure 3. It can be seen from the figure that the thermal transition temperatures of ASC, ALSC, and PSC are 106.41 °C, 103.76 °C, and 106.58 °C, respectively. The transition temperature of PSC and ASC is higher, while that of ALSC is obviously lower. The results indicate that the thermal stability of PSC is the best, that of ALSC is the worst, and that the difference between ASC and PSC is not obvious.

### 3.4. Microstructure

Ultrastructure is an important factor in evaluating potential applications of collagen [17]. Macroscopic observation showed that ASC, ALSC, and PSC were different in appearance and structure (Figure 4). From the SEM images of ASC, ALSC, and PSC, it can be seen that ASC has a dense and uneven block-like structure, which may be related to collagen moisture absorption. However, both ALSC and PSC have loose, porous sponge-like structures. Further magnification of 300 times revealed that the microstructure of PSC has higher porosity and larger pore size and is more neatly arranged, more like a cave; while the pores of ALSC are irregular, small, and interconnected. These results showed that different extraction methods had obvious effects on the microstructure of chicken bone collagen.

### 3.5. Processing Properties

#### 3.5.1. Solubility

The effects of pH and NaCl on the solubility of ASC, ALSC, and PSC are shown in Figure 5. ASC, ALSC, and PSC have higher solubility in the low pH range (1 to 3); especially when the pH is closer to 2, the collagen solubility is the highest. When the solution pH was higher than 4, the collagen solubility decreased sharply. Collagen had the lowest solubility when the pH was around 6–7; specifically, ALSC had the lowest solubility at pH 6 and ASC and PSC had the lowest solubility at pH 7. Then with the increase in pH, the solubility of collagen gradually increased. In addition, the solubility of ASC, ALSC, and PSC decreased gradually with the increase in NaCl concentration. Overall, there was little difference in NaCl sensitivity among the three collagens.

#### 3.5.2. Turbidity

Turbidity is a method of monitoring protein aggregation. As shown in Table 4, ALSC had the highest turbidity, and the turbidity of ASC and PSC was significantly lower than that of ALSC (*p* < 0.05), while there was no significant difference in the turbidity of ASC and PSC (*p* > 0.05). Generally, turbidity directly reflects the quality of protein samples. The reason for the difference of collagen turbidity may be the different size of collagen molecules extracted by different methods [26,43].

#### 3.5.3. WHC and OAC

WHC is one of the important functional properties of proteins; it is related to protein texture and fluctuates under the influence of factors, including ionic strength, protein concentration, and binding capacity [44]. As shown in Table 4, there were significant differences in the WHC of ASC, ALSC, and PSC, where ASC was significantly lower than ALSC and PSC, and ALSC was significantly lower than PSC (*p* < 0.05), which was similar to the results reported by Akram [24]. Therefore, PSC may have better application prospects.

As an important characteristic of protein, OAC is related to product texture and flavor. The assay showed that there were significant differences (*p* < 0.05) in OAC among ASC, ALSC, and PSC, which were 4.98, 6.37, and 8.17 mL/g, respectively. This value is lower than the previous report of chicken bone collagen [24] but better than the collagen extracted from aquatic products [45].

#### 3.5.4. Emulsifying and Foaming

Foaming is an important functional property of protein, which is generally measured by the foaming ability (FA) and foaming stability (FS). As shown in Table 4, the FA values of ASC, ALSC, and PSC were 109.78%, 113.27%, and 124.83%, respectively, of which PSC was significantly different from ASC and ALSC (*p* < 0.05), and ALSC was not significantly different from ASC (*p* > 0.05); The FS values were 103.83%, 105.89%, and 108.78%, respectively, and there were significant differences among them (*p* < 0.05).

Emulsification is the ability of protein to form emulsions with water, which is usually evaluated by emulsification activity index (EAI) and emulsification stability index (ESI). The ESI values of ASC, ALSC, and PSC were 50.65 m^2^/g, 49.22 m^2^/g, and 66.14 m^2^/g, respectively. PSC was significantly higher than ALSC and ASC (*p* < 0.05), but there was no significant difference between ALSC and ASC (*p* > 0.05). The EAI values were 51.10 min, 42.50 min, and 66.69 min, respectively. PSC was significantly different from ASC and ALSC (*p* < 0.05), but there was no significant difference between ASC and ALSC (*p* > 0.05).

### 3.6. In Vitro Antioxidant Properties of Collagen

DPPH•, HO•, O_2_^−^•, and ABTS^+^• are common radicals that participate in biological oxidation leading to oxidative stress, which destroy macromolecules such as carbohydrates, nucleic acids, lipids, and proteins [30,46]. As shown in Figure 6, ASC, ALSC, and PSC all had the ability to scavenge DPPH•, HO•, O_2_^−^•, and ABTS^+^• radicals, and the scavenging activity was dose-dependent, and chicken bone collagen scavenged activity order of DPPH•, HO•, O_2_^−^•, and ABTS^+^ radicals is PSC > ASC >ALSC. The EC_50_ of ASC, ALSC, and PSC for the four radicals were as follows: EC_50_ (of DPPH) were 2.80, 3.50, 2.41 mg/mL; EC_50_ (of HO•) were 4.05, 5.55, 3.71 mg/mL, respectively; EC50 (of O_2_^−^•) were 3.60, 4.47, 3.42 mg/mL; EC_50_ (of ABTS^+^•) were 4.52, 4.97, 4.02 mg/mL, respectively. It can also be seen from the EC_50_ value that the in vitro antioxidant activity of chicken collagen is PSC > ASC > ALSC.

### 3.7. Color and Sensory Evaluation

As shown in Table 5, the brightness values (*L**) and yellowness values (*b**) were significantly different among ASC, ALSC, and PSC (*p* < 0.05); the brightness value ASC > PSC > ALSC, and the yellowness value ALSC > PSC > ASC, while redness values (*a**) were not significantly different (*p* > 0.05). The color of PSC was similar to that of chicken foot collagen extracted by alkaline protease [47]. Figure 7 shows the color difference between ASC, ALSC, and PSC more intuitively. Therefore, in terms of color, ASC and PSC are more acceptable than ALSC.

From the appearance of the sample, ASC is a pure white powder with poor spatial structure, while PSC is not as bright as ASC, but PSC has a sponge-like spatial structure. In contrast, ALSC is yellowish and has poor brightness. Although it has a reticular spatial structure, the mesh is large and non-uniform, and the texture is hard (Figure 7).

Figure 8 shows the sensory evaluation results for ASC, ALSC, and PSC. Sensory panel members point out that the three collagen samples differed obviously in color, transparency, viscosity, and odor. Furthermore, they point out that PSC has a stronger aroma and low odor than ALSC and ASC; PSC and ASC have no obvious difference in color, transparency, or viscosity; ALSC was significantly different from PSC in all sensory indices except viscosity, while ASC was significantly different from PSC in flavor (*p* < 0.05). ALSC and PSC have significant differences in all sensory indicators except for viscosity, and, in ASC and PSC, only flavor shows significant differences (*p* < 0.05). Similarly, Nik et al. [48] also reported that there were significant differences in sensory characteristics of duck foot gelatin extracted with different types of acids.

## 4. Discussion

### 4.1. Effects of Extraction Methods on Collagen Characteristics

The differences in the chemical composition of chicken bone meal and collagen in this experiment in comparison with the previous results [32,49,50] may be caused by the differences in the breed, age, nutritional level of raw chicken bone, and extraction method. The high yield of PSC may be due to the specific cleavage of peptides by proteases, facilitating the extraction of collagen from the fibrillary matrix. The amino acid composition of ASC, ALSC, and PSC showed that they all had typical collagen characteristics. Amino acid composition and content affect the stability of collagen triple-helix structure. The content of Gly in chicken bone collagen is close to 1/3 of the total amino acid content. Gly contributes to the formation of collagen superhelix, makes collagen α chain tightly together, and endows collagen with stable triple-helix structure [51]. In addition, hydroxyproline content has been proved to play an important role in stabilizing the triple-helix structure of collagen [17]. Therefore, PSC has the highest thermal stability, which is consistent with DSC results.

According to the spectral characteristics of collagen, the triple helical structure of collagen has a maximum absorption peak between 210 nm and 240 nm, so UV full spectrum scanning is often used to evaluate the structural integrity and purity of collagen [17]. In this test, PSC has a maximum absorption peak at 215.77 nm, which indicates that PSC has a more complete triple-helix structure or higher purity. However, the maximum absorption wavelength of ASC and ALSC was lower than 210 nm, which may be due to the influence of acid and alkali on collagen structure and amino acid composition or interference from impurities. The FTIR of ASC, ALSC, and PSC are similar to those of collagen extracted from chicken or fish previously reported [17,24,33], indicating that they have the triple-helix structure of collagen. However, there were differences in FTIR spectra among the three collagens, indicating differences in the secondary structure of the three collagens, especially the strong absorption of ASC and ALSC near 1030 cm^−1^, which may be due to the effect of acids and alkali on collagen structure or the effect of impurities. In brief, FTIR spectra showed that ASC, PSC, and ALSC all have the characteristic triple-helix structure of collagen, but acid and alkali extraction may have adverse effects on the secondary structure of chicken bone collagen, and the triple-helix structure of PSC is more stable.

The difference between the ultrastructure of ASC and previous reports may be due to the slight moisture absorption of ASC. The ultrastructure of ALSC and PSC are similar to those of bovine skin gelatin [26], chicken bone collagen [52], and chicken bone gelatin [32] but different from ultrasound-assisted extraction of chicken breast cartilage collagen [24], which may be attributed to the effect of ultrasound. Previous studies have shown that collagen with fibrous and lamellar membrane structure can be used for tissue formation, growth, wound healing, gene expression, etc. As a drug carrier, a more uniform and regular structure is more conducive to the uniform distribution of drugs [17]. Therefore, in terms of microstructure, PSC with a uniform and porous sponge structure may be more suitable for functional food or medical research. Thermal stability is one of the main indices that determines the application range of collagen. The results of thermal transition temperatures of chicken bone collagen prepared in this test are similar to those of deer, pig, and bovine bone collagen determined by Yu [21] but different from those reported by others [24], possibly due to the high-temperature pretreatment of raw materials. Many studies have reported that collagen peptides prepared by high-temperature pretreatment still have high antioxidant properties [21,22]; for example, high temperature and pressure are also used to prepare active peptides [53]. The thermal stability of PSC is the best, while that of ALSC is the worst, which may be related to amino acid difference in addition to the effect of alkali on collagen structure.

### 4.2. Effect of Extraction Methods on the Processing Properties of Collagen

The solubility of collagen is related to its isoelectric point, which is reported to be between 6 and 9 [18]. The dissolution results of collagen prepared in this study under different pH were similar to those of collagen extracted from the skin of channel catfish [18] and red snapper [54]. The reason why NaCl reduces the solubility of collagen may be that the increased ionic strength changes the interaction between protein chains or because salt ions compete for water with proteins by increasing hydrophobic interaction and aggregation. In this study, PSC has higher solubility than ASC and ALSC, which is consistent with collagen extracted from catfish skin by different methods [18,55]. The difference of collagen structure, activity, and molecular properties caused by different extraction methods may be the reason for the different solubility of chicken bone collagen.

The differences in WHC and OAC among ASC, ALSC, and PSC may be related to their surface hydrophobicity, different charged perssad, and structural differences. The reason for the difference between the measured values and previous reports [24] may be related to the extraction method and conditions, raw materials, collagen type, oil type, etc. As an important functional characteristic of collagen, emulsification is affected by pH, ion concentration, temperature, and other factors, while foaming is affected by sugars, salts, lipids, and other components and factors. The foaming and emulsifying properties of PSC, ASC, and ALSC are different and are different from those reported by Zhou [47] and Akram [24], which may be due to differences in extraction methods, raw materials, collagen molecular weight, and ionic strength.

### 4.3. Effect of Extraction Methods on Activity of Collagen In Vitro

Biological activity is the premise that determines the application of collagen in food, medicine, cosmetics, and other fields. The antioxidant activities of the three chicken bone collagens extracted in this study were dose-dependent, and the extraction method had a significant effect on the activity of collagen. Similarly, the free radical scavenging activity of the enzyme-soluble collagen extracted from the swim bladder was stronger than that of the acid-soluble collagen extracted from the same raw material [17]; the antioxidant activity of collagen is related to the connection between peptide bonds and amino acid residues and to the affinity of amino acid side chains between free radicals. Therefore, the stable triple helical structure and amino acid differences of chicken bone collagen may be the reason why its antioxidant activity is not as strong as that of collagen or collagen peptides from other sources [30,46,56]. However, the reason for the difference in antioxidant activity among ASC, ALSC, and PSC may be that different extraction methods lead to different molecular weight, hydrophobicity, aromatic amino acid residues, and amino acid sequences of collagen [57].

### 4.4. Effects of Extraction Methods on Sensory Quality of Collagen

Sensory quality is the most intuitive indicator for evaluating the overall commercial quality of collagen. Collagen is a white, odorless, transparent powder. In this experiment, the brightness of ASC and PSC was the highest, while the color of ALSC was yellow, which may be due to the amide bonds rich in collagen molecules and the oxidation of -NH_2_ under alkaline conditions. PSC has the aromatic flavor of an enzyme, the best sensory evaluation, and a uniform spongy spatial structure. Although ALSC also has a spongy structure, the texture is hard, and the space structure is disordered. Therefore, enzymatic extraction is one of the best methods to prepare chicken bone collagen.

## 5. Conclusions

In this study, three kinds of collagen (ASC, ALSC, and PSC) were extracted from discarded chicken bones, and the effects of different extraction methods on the characteristics, physical and chemical properties, functional properties, and sensory quality of chicken bone collagen were analyzed by amino acid analysis, SDS-PAGE, FTIR, DSC, UV full spectrum scanning, SEM, and other methods. Although the collagen extracted by the three methods had the typical triple-helix structure and amino acid composition characteristics of collagen, acid and alkali extraction seemed to have adverse effects on the collagen secondary structure and microstructures. In addition, PSC had better antioxidant activity, thermal stability, and sensory acceptability. Therefore, we concluded that chicken bone collage prepared by enzymatic hydrolysis has a wider application prospect in food and medicine due to its complete and stable structure, higher yield, thermal stability, processability, antioxidant activity, and sensory acceptability. However, in terms of extract methods, it is still necessary to assist collagen extraction with more efficient, energy-saving, environmental protection, sustainable, and innovative new technologies in the future to meet the demand of the collagen market. In addition, we prepared and compared the effects of different extraction methods on collagen from discarded chicken bone, which provides an effective method for the high-value comprehensive utilization of waste chicken bone by-products.

## Figures and Tables

**Figure 1 foods-12-00202-f001:**
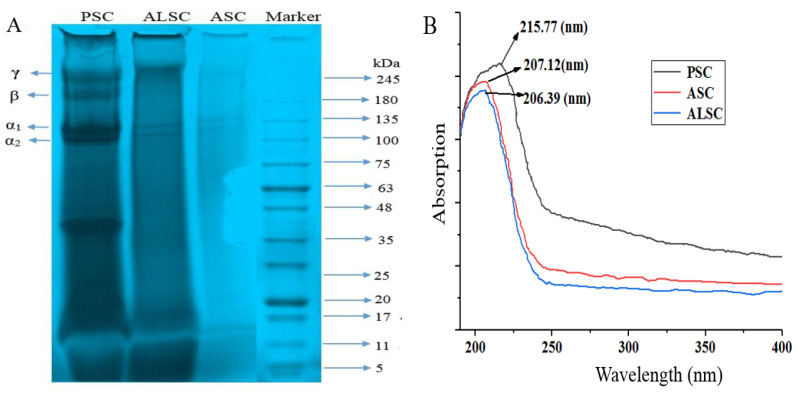
SDS-PAGE electrophoresis (**A**) and UV spectra (**B**) of ASC, ALSC, and PSC. Note: Since the UV scanning instrument can not export the original image, GetData software was used to take points and draw Figure 1B.

**Figure 2 foods-12-00202-f002:**
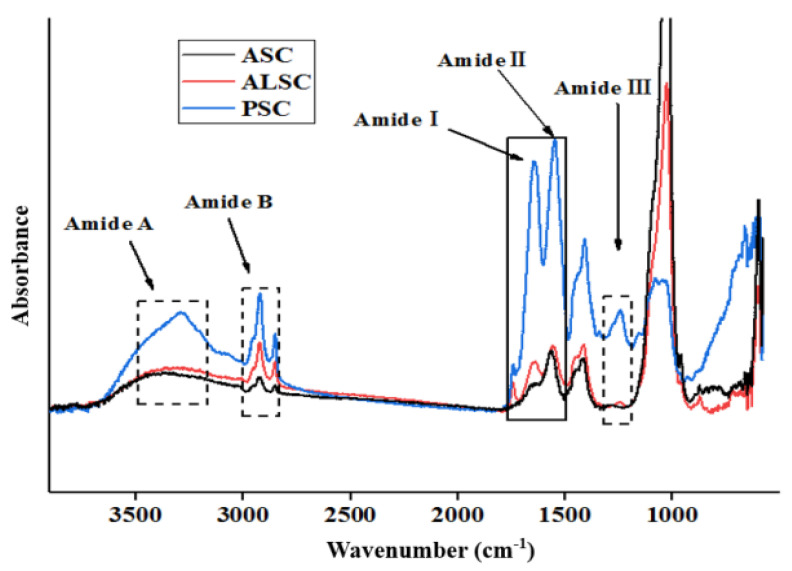
Fourier transform infrared spectra of ASC, ALSC, and PSC.

**Figure 3 foods-12-00202-f003:**
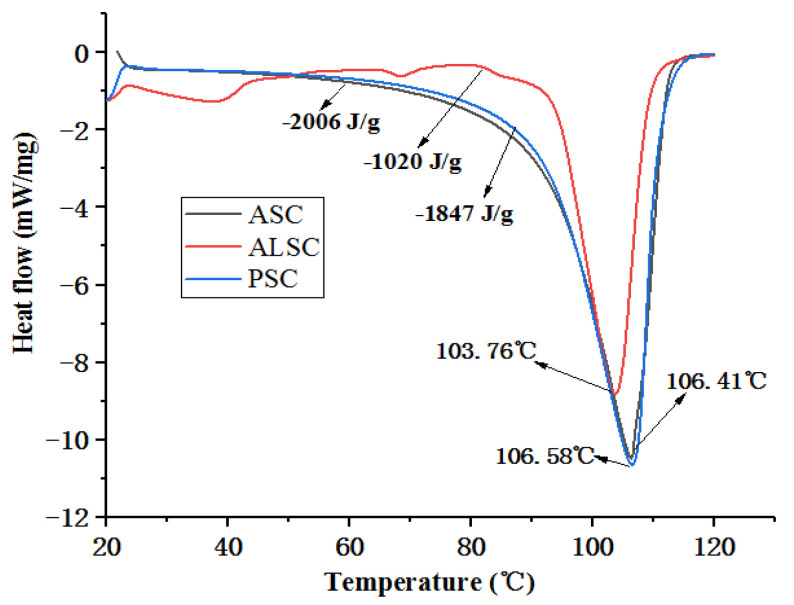
Differential scanning calorimetry of ASC, ALSC and PSC.

**Figure 4 foods-12-00202-f004:**
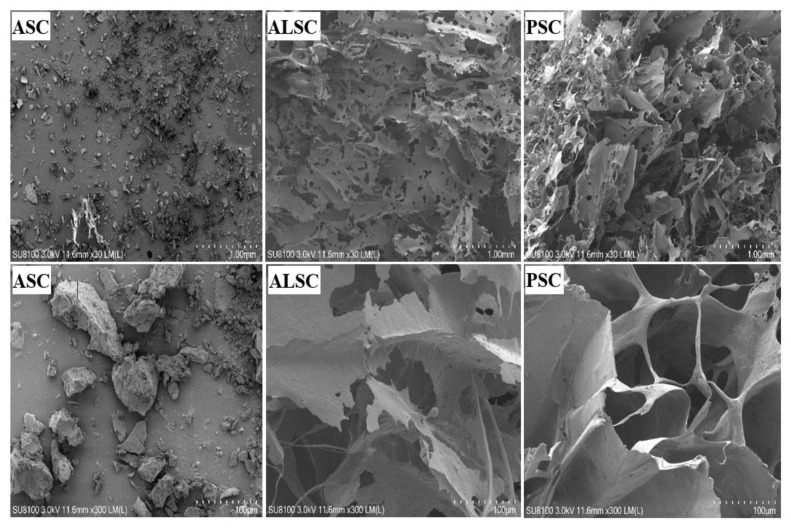
SEM microscopic structure of ASC, ALSC, and PSC (upper row ×30, lower row ×300).

**Figure 5 foods-12-00202-f005:**
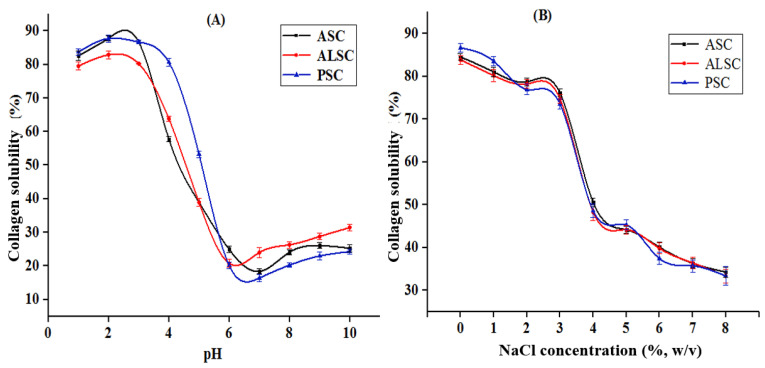
Effects of pH (**A**) and NaCl (**B**) on the solubility of ASC, ALSC, and PSC.

**Figure 6 foods-12-00202-f006:**
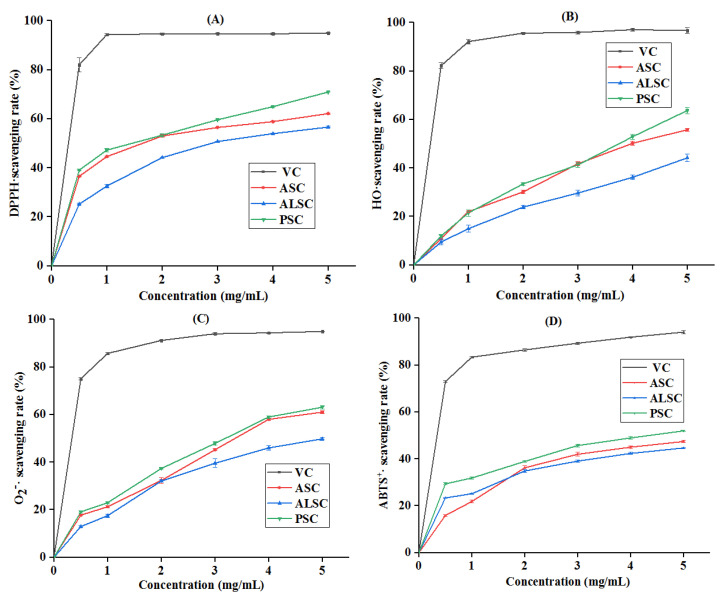
DPPH•(**A**), HO•(**B**), O_2_^−^•(**C**), and ABTS^+^•(**D**) scavenging activities of ASC, ALSC, and PSC.

**Figure 7 foods-12-00202-f007:**
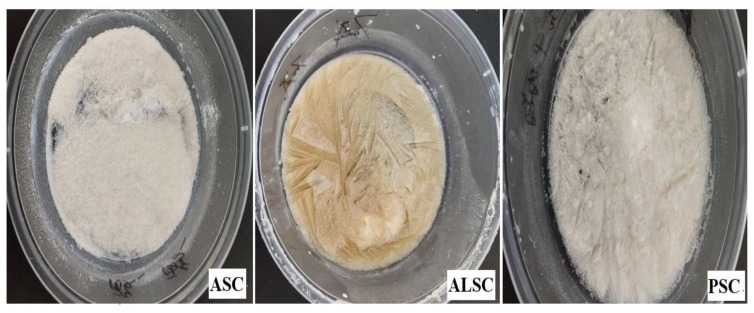
Effect of extraction method on the overall appearance of chicken bone collagen.

**Figure 8 foods-12-00202-f008:**
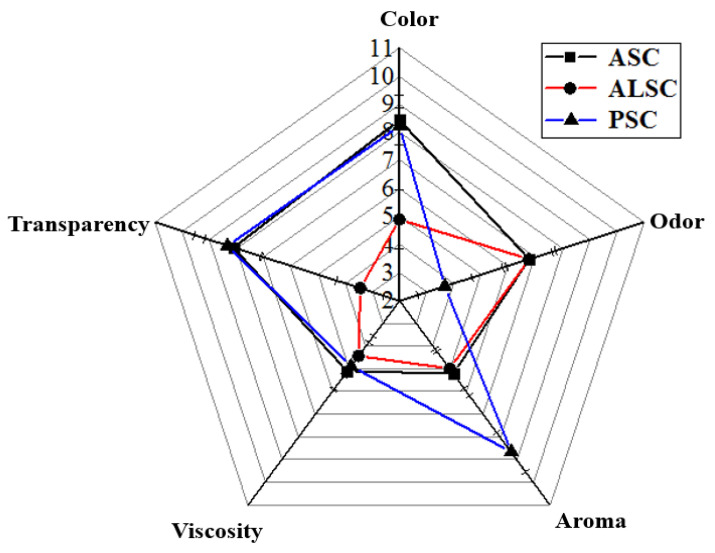
Radar chart of sensory analysis of ASC, ALSC, and PSC.

**Table 1 foods-12-00202-t001:** Composition of chicken bone meal, extraction rate, and composition of collagen.

Index	Chemical Composition of Raw Materials and Collagen			
	Bone Meal	ASC	ALSC	PSC
Yield (%)	-	3.39 ± 0.37 ^b^	2.42 ± 0.12 ^c^	9.63 ± 0.60 ^a^
Moisture (%)	64.80 ± 1.73 ^a^	5.10 ± 0.48 ^b^	4.96 ± 0.28 ^b^	5.18 ± 0.30 ^b^
Ash (%)	7.92 ± 0.36 ^a^	1.47 ± 0.12 ^b^	1.44 ± 0.11 ^b^	1.43 ± 0.08 ^b^
Protein (%)	15.79 ± 0.55 ^c^	83.37 ± 0.72 ^b^	81.37 ± 2.11 ^b^	87.63 ± 2.36 ^a^
Fat (%)	7.45 ± 0.78 ^a^	0.43 ± 0.02 ^c^	1.09 ± 0.03 ^b^	0.20 ± 0.01 ^d^

Note: Data are expressed as mean ± SD (*n* = 3); Different right shoulder letters in peer data indicate significant differences (*p* < 0.05), the same as below.

**Table 2 foods-12-00202-t002:** Amino acid compositions of chicken bone collagen.

Amino Acid Species	Content (%, g per 100 g of Protein)		
	ASC	ALSC	PSC
Asp	4.56	6.30	4.45
Thr	2.53	3.37	2.37
Ser	3.81	4.63	3.72
Glu	9.28	10.06	9.14
Gly	26.74	22.67	27.86
Ala	9.83	9.12	9.61
Val	1.40	1.97	1.45
Met	1.84	2.72	2.11
Ile	1.35	2.54	1.48
Leu	2.81	4.72	2.54
Tyr	0.87	0.93	0.49
Phe	2.44	3.71	2.03
Lys	3.34	3.96	3.15
His	0.48	0.88	0.57
Arg	5.17	5.28	5.22
Pro	10.55	8.27	11.60
Hyp	9.51	6.29	9.83

**Table 3 foods-12-00202-t003:** FTIR peak locations and assignment for ASC, ALSC, and PSC.

Properties	Peak Wave Number (cm^−1^)			Assignment	References
	ASC	ALSC	PSC		
Amide A	3340.91	3371.93	3290.62	N-H stretch and hydrogen bond	[37,42]
Amide B	2923.13	2923.51	2921.42	CH_2_ asymmetric stretch	
-	2852.32	2856.20	2851.19	CH_2_ symmetric stretch	
Amide I	1644.51	1652.18	1644.4	C=O stretch/hydrogen bond coupled with COO-	
Amide II	1559.72	1560.49	1548.3	NH bends and CN stretch	
Amide III	1244.24	1250.84	1243.22	NH bends and CN stretch	
-	1025.72	1027.95	1080.46	C-O stretch	

**Table 4 foods-12-00202-t004:** Effects of extraction methods on the functional properties of chicken bone collagen.

Processing Properties	ASC	ALSC	PSC
Turbidity (mg/L)	47.28 ± 5.58 ^b^	132.94 ± 26.13 ^a^	57.67 ± 6.73 ^b^
WHC (g/g)	0.71 ± 0.03 ^c^	0.85 ± 0.01 ^b^	1.06 ± 0.04 ^a^
OAC (mL/g)	4.98 ± 0.51 ^c^	6.35 ± 0.43 ^b^	8.17 ± 0.93 ^a^
FA (%)	109.78 ± 1.44 ^b^	113.27 ± 1.07 ^b^	124.83 ± 2.67 ^a^
FS (%)	103.83 ± 0.60 ^c^	105.89 ± 1.40 ^b^	108.78 ± 0.82 ^a^
ESI (m^2^/g)	50.65 ± 5.44 ^b^	49.22 ± 3.97 ^b^	66.14 ± 6.82 ^a^
EAI (min)	51.10 ± 4.50 ^b^	42.50 ± 4.98 ^bc^	66.69 ± 5.72 ^a^

Note: Different right shoulder letters in peer data indicate significant differences (*p* < 0.05).

**Table 5 foods-12-00202-t005:** Color of chicken collagen extracted by acid, alkali, and enzyme.

Collagen	*L**	*a**	*b**
ASC	93.35 ± 1.22 ^a^	0.76 ± 0.46	0.55 ± 0.42 ^c^
ALSC	62.38 ± 1.12 ^c^	0.91 ± 0.57	20.93 ± 1.85 ^a^
PSC	81.24 ± 0.68 ^b^	0.24 ± 0.17	11.84 ± 0.52 ^b^

Note: Different right shoulder letters in column data indicate significant differences (*p* < 0.05).

## Data Availability

Data is contained within the article.

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
