# Peer review of "Effects of Extraction Methods on the Characteristics, Physicochemical Properties and Sensory Quality of Collagen from Spent-Hens Bones"

_foods, 2023, doi:10.3390/foods12010202_

Round 1

Reviewer 1 Report

The authors have compared three collagen extraction process. I think that the results have been exposed with clarity and the conclusions are according with them. I consider that the statistical data treatment gives the manuscript a high value.

In my opinion, the study is interesting and the manuscript has been correctly written and therefore I considered that the manuscript can be accepted with minor changes.

I consider that the authors should change the abstract, because the redacted abstract it's identical to the described conclusions in the manuscript.

Author Response

Dear reviewer, first of all thank you for reviewing my manuscript and for your valuable comments. Your review comments and serious attitude towards academics have benefited me a lot. Because of my carelessness caused many problems, I sincerely apologize to you. Secondly, I have carefully revised and supplemented the corresponding questions in accordance with your review requirements. The changed part has been marked in red font. Please check it.

Reviewer 2 Report

There some recommendations how to improve the paper.

The title: typing error …”collagenfromspent…”

Abstract, page 26: A technique for sample analysis is “scanning electron microscopy” (SEM); not “microscope”. The same in line 153 etc.

Abstract, page 27: typing error “…microstructure.In addition…”

Keywords: instead of “Chicken bone” use plural (”Chicken bones”).

Page 2, line 87: Can the authors specify the age of spent-hens?

Page 5, line 188: typing error in the heading

Page 7, line 273: I do not fully understand the term “functional perssad in collagen”. Does it refers to functional groups in collagen ?

Chapter 3.2.: Check the font size of the text and typing errors. Why word “radical” in the text is underlined?

I am wondering that it would be very helpful give the FOODS journal readers some detailed examples of applications of prepared chicken bones collagens with respect to their functional properties (as stated in Table 4). There is no huge difference among ASC, ALSC and PSC. Nevertheless, the is huge difference between all 3 collagens to hold water (WHC) in contrast with holding oil (OAC). Some food applications require high WHC, while the others better OAC… In discussion part it should be reflected.

Generally, there are typing errors throughout the paper. Check it properly.

Author Response

(The authors gave the same response as above.)

Reviewer 3 Report

The work described by Cao and colleagues describes the impact of three different extraction methods on the properties of the obtained collagen. The work is interesting and the collagen extracts are well characterized. However, the results are not enhanced by the discussion, which is often limited to saying "the X value is greater than the Y value..." and so on. Hence, the discussion part should be improved by highlighting the significance of the findings and possible outcomes of the research.

General

English must be revised throughout the document; many sentences contain errors and some sentences are difficult to understand (for example, lines 44-46: the sentence seems unfinished)

Line 3: “collagenfromspenthens”, Line 22: “characteristicsand”, Line 48: “solvedurgently”, Line 66: “materials.Animal”, and others in the document: insert spaces

Materials and methods

Lines 85-86: the drying parameters (temperature, time, type of drying) must be indicated. The crushing parameters (tool used, settings, granulometry) must also be reported. All of this information is important as it can influence the extraction process and the characteristics of the collagen.

Lines 91-93: All abbreviations must be defined when first mentioned

Lines 98-100: The pH of the suspension must be reported as it can influence the enzymatic activity. Why was 4°C used as the extraction temperature? The optimal temperature of pepsin is 37°C.

Lines 107-112: the pH must also be reported for acid and alkaline extractions.

Paragraph 2.3.2 Yield: the yield should also be measured by dividing the amount of collagen extracted by the amount of total collagen initially present in the starting bones, in order to have an idea of the efficiency of the process and how much collagen is still left in the bone.

Results and discussion

Lines 244-246: No discussion can be made without knowing the statistical significance of the observed differences

Lines 348-349: Is "significant" supported by the statistic?

Tables

Table 1: to facilitate the comparison (especially with raw bones) data should be reported on dry matter

Table 2: why was the statistic not performed for amino acids, as it was done for Table 1? What is the unit of measurement? g of amino acid / 100 g of sample? Must be specified.

Figures

Figure 1: Any guesses of band smearing?

Author Response

(The authors gave the same response as above.)

Round 2

Reviewer 3 Report

Thanks for taking my suggestions into consideration. However, many points have not been resolved.

The discussion part was not enhanced by highlighting the significance of the results and possible outcomes of the research.

The activity of pepsin at 4°C must be measured and controlled, since it is very far from its optimal temperature, otherwise it makes no sense to add it if its specific activity is not known.

The authors state that there is no need to measure the pH for acidic and alkaline extractions, as it can be calculated. This is true, but proteins can have a buffering effect, so it is not always certain that the pH of the mixture will coincide with the calculated one.

The amount of protein (presumably mainly collagen) in the starting material can be measured quickly with Kjeldahl or Dumas. This data should be provided. No extraction yield can be measured otherwise.

Statistics should also be provided, if the authors have the raw data some simple tests can be done with the different software available.

Table 2: I'm not sure I understand why statistics can't be done here, the data is single measurements? In this case, at least a second replica should be done.

Author Response

Dear reviewer:

Thank you for reviewing my manuscript again. In response to your comments, we would like to make the following detailed explanations:

  1. The discussion part was not enhanced by highlighting the significance of the results and possible outcomes of the research.

Dear reviewer, thank you very much for your comments. If possible, please specify to us which part of the discussion needs to be improved so that we can make corresponding modifications.

  1. The activity of pepsin at 4°C must be measured and controlled, since it is very far from its optimal temperature, otherwise it makes no sense to add it if its specific activity is not known.

Dear reviewer, thank you for your opinion. For this problem, you asked me why I chose 4℃ extraction before, and I gave you the reasons: ((1) Collagen has poor thermal stability, and low temperature extraction can avoid its failure or denaturation due to temperature increase. (2) Glacial acetic acid has high volatility, low boiling point, and low temperature extraction can avoid the influence of acid volatilization on the concentration of extraction solution. (3) Protein at 4℃ is the most stable, the best extraction effect, but also convenient operation). All the literatures related to collagen extraction were extracted at 4℃, but 95% of the studies did not measure the activity of enzyme at 4℃. There are too many literatures to support the reliability of my research method (such as doi:10.3390/md16050161, doi:10.3390/md17020078: Marine drugs; doi:10.1016/j.foodchem.2019.125544: food chemistry, and so on). The main objective of this study was to investigate the effect of extraction methods on collagen properties rather than enzyme activity. Although the reviewer's opinion is reasonable to some extent, we believe that this experiment does not need to measure enzyme activity at 4℃. Please consider our views.

  1. The authors state that there is no need to measure the pH for acidic and alkaline extractions, as it can be calculated. This is true, but proteins can have a buffering effect, so it is not always certain that the pH of the mixture will coincide with the calculated one.

Dear reviewer, Thank you for your advice, according to yours requirements, we added the solution pH measured before. Please check lines 101 and 112.

  1. The amount of protein (presumably mainly collagen) in the starting material can be measured quickly with Kjeldahl or Dumas. This data should be provided. No extraction yield can be measured otherwise.

Dear reviewer, Thank you for your advice, we have determined the crude protein content in raw materials by Kjeldahl in our original manuscript, and it has been listed in Table 1, please check.

  1. Statistics should also be provided, if the authors have the raw data some simple tests can be done with the different software available.

Dear reviewer, we do not understand your meaning of this question. I don't know whether you wants us to provide the original data or our statistical method is wrong? I think there is no problem with our statistical method, and our original data has also been submitted to the doctoral supervisor.

  1. Table 2: I'm not sure I understand why statistics can't be done here, the data is single measurements? In this case, at least a second replica should be done.

Dear reviewer, this question has been explained to the reviewer in my previous reply, and listed some cases in which similar studies were not repeated in the determination of collagen amino acids (e.g: doi:10.3390/md16050161; Doi: 10.3390/md17020078; Doi: 10.1016/j.i jbiomac 2017.07.013). We admit that this is because we referred to these literatures in the experimental design, which caused any inaccuracy. Please forgive our shortcomings.
